# Genomic Immune Evasion: Diagnostic and Therapeutic Opportunities in Head and Neck Squamous Cell Carcinomas

**DOI:** 10.3390/jcm11247259

**Published:** 2022-12-07

**Authors:** Kedar Kirtane, Maie St. John, Harry Fuentes-Bayne, Sandip P. Patel, Armen Mardiros, Han Xu, Eric W. Ng, William Y. Go, Deborah J. Wong, John B. Sunwoo, John S. Welch

**Affiliations:** 1Moffitt Cancer Center, Tampa, FL 33612, USA; 2Otolaryngology, UCLA School of Medicine, Los Angeles, CA 90095, USA; 3Mayo Clinic, Rochester, MN 55902, USA; 4Moores Cancer Center, UCSD School of Medicine, San Diego, CA 92093, USA; 5A2 Biotherapeutics, Agoura Hills, CA 91301, USA; 6Otolaryngology, Stanford University, Palo Alto, CA 94305, USA

**Keywords:** HNSCC, immune evasion, loss of heterozygosity (LOH), HLA, immuno-oncology, biomarkers, head and neck cancers, therapeutic targets

## Abstract

Head and neck squamous cell cancers (HNSCCs) represent a diverse group of tumors emerging within different mucosal surfaces of the oral cavity, nasopharynx, oropharynx, larynx, and hypopharynx. HNSCCs share common clinical risk factors and genomic features, including smoking, alcohol, age, male sex, aneuploidy, and *TP53* mutations. Viral initiating and contributing events are increasingly recognized in HNSCCs. While both Epstein–Barr Virus (EBV) and human papilloma virus (HPV) are observed, EBV is more frequently associated with nasopharyngeal cancers whereas HPV is associated with oropharyngeal cancers. HNSCCs are associated with high tumor mutational burden and loss of tumor suppressor gene function, especially in *TP53* and X-linked genes. Multiple lines of evidence suggest that HNSCCs are subject to immunologic surveillance and immune-induced evolutionary pressure that correlate with negative clinical outcomes. This review will discuss genomic mechanisms related to immune-mediated pressures and propose prognostic and therapeutic implications of detectable immune escape mechanisms that drive tumorigenesis and disease progression.

## 1. Introduction

Head and neck squamous cell carcinomas (HNSCCs) represent the most common histologic subtype within the broader and highly heterogeneous category of head and neck tumors [1]. HNSCC is the seventh most common cancer worldwide and accounts for ~4% of all cancers in the United States, with an estimated 66,000 new cases diagnosed each year. Globally, more than 500,000 new cases and 250,000 deaths attributed to HNSCC occur each year [2,3]. Diagnosis of HNSCC has long been associated with smoking, alcohol, exposure to environmental carcinogens, age, male sex, and unique viral etiologies [4]. Although early-stage tumors are often associated with good prognosis, most patients present with locally advanced disease, of whom about 50% will relapse within 1 year. Median survival rates remain modest, with 5 year survivals ranging from50 to 60% in the era before immunotherapy [5,6]; current immunotherapy approaches demonstrate significant albeit modest gains of 2–4 months in the relapsed and refractory setting [7]. Patients who do achieve a cure from the disease often suffer consequences of complicated and often morbid surgeries or long-term sequelae from chemotherapy and radiation to the head and neck, resulting in temporary or permanently altered eating, speech, and quality of life [8]. Among cancer survivors, those with HNSCC have the second highest rate of suicide after those with pancreatic cancer, which is attributed to psychological stress and compromised quality of life [9]. New diagnostic tools and therapeutic strategies are needed to inform patient care, provide meaningful improvement to progression-free and overall survival, and minimize long-term toxicity of therapy.

## 2. Prognostics in Head and Neck Squamous Cell Carcinomas 

Outcomes in HNSCC remain inadequate, particularly in the metastatic and relapsed setting. Thus, there is a tremendous need to identify molecular biomarkers that provide prognostic and therapeutic opportunities, which may identify subgroups of patients who could benefit from novel approaches based on clinical or molecular characteristics.

Diverse clinical and molecular strategies have been explored to identify HNSCC biomarkers yielding an estimate of more than 100 different candidates [10,11]. Although many of these have shown significant correlations with diagnosis and prognosis, few have been prospectively validated in clinical trials or for use in clinical practice to inform specific therapeutic strategies. The most prominent and reproducible clinical and pathologic markers include age, smoking status, positive margins, tumor depth, and extranodal disease [2,10,12]. Recently, multiple immune signatures have been associated with prognosis, including simple peripheral blood ratios of neutrophil-to-lymphocyte, which were found to correlate with poorer overall survival in a pooled analysis of 929 patients from 14 studies (hazard ratio [HR], 2.03; 95% CI, 1.50–2.74) [2,6,10,13,14,15,16]. The tumor microenvironment (TME) of NHSCCs is also enriched in non-T immune cells, which harbor chronic inflammation induced via expression of pro-inflammatory and pro-angiogenic cytokines that recruit suppressive immune cells and promote hypoxic conditions associated with poor outcomes [17]. For example, serum IL-6 concentrations were found to be higher in patients with HNSCC compared to healthy subjects and patients with premalignant lesions [18]. Several studies reported that salivary IL-6 concentrations are significantly increased in patients with oral and oropharyngeal diseases, and higher IL-6 expression is inversely related to prognosis and survival of patients with oral squamous cell carcinomas [19]. Novel molecular testing modalities being explored include evaluation of the oral microbiome, which often impacts mucosal tissue and has been shown to contribute to carcinogenesis in up to 15% of oral cancers. In addition, measuring circulating tumor DNA [20,21,22] appears to correlate with aggressive features and increased metastatic potential. Many of these approaches appear promising, but additional research is required to identify clinically meaningful biomarkers that have sufficient specificity, reasonable cost, and ease of application to inform treatment decisions in a broad population of patients. In the last decade, advances in the next-generation sequencing (NGS) of tumor samples have provided a sensitive, high-throughput sequencing technology that is being broadly integrated into tumor diagnostics. NGS enables the simultaneous measurement of multiple prognostic features involving recurrent patterns of mutations and genomic deletions or amplifications. The capture and sequencing of curated genome regions enables detection of a broad spectrum of DNA variations associated with tumorigenesis and enables detection of diagnostic, potentially therapeutically relevant targets [23].

The heterogeneity of HNSCC may enable the identification of molecular and diagnostic markers to help define sub-populations with prognostic or therapeutic relevance [24]. Diverse immune escape mechanisms may also provide opportunities for patient-directed, targeted therapy. However, the extent of diversity within HNSCC patient populations has been daunting. Inconsistent strategies have been applied to evaluate and characterize patient populations and compare treatment approaches, sometimes even within the same therapeutic class. Future studies will be required to define germane subsets within patients with HNSCC, and validated biomarkers are likely to be therapy specific.

## 3. Risk Factors for Head and Neck Squamous Cell Carcinomas: Viruses and Sex

Within non-nasopharyngeal cancers, two distinct forms of HNSCC have emerged with different risk factors: tobacco and alcohol exposure, and human papillomavirus (HPV) infection [25,26]. Patients with HPV-negative HNSCC present more frequently with HNSCC affecting the oral cavity and larynx and have prolonged exposure of this anatomy to carcinogens via smoking and alcohol consumption. Generally, patients with HPV-positive HNSCC present with oropharyngeal origins and have more favorable outcomes in response to chemotherapy and radiotherapy [27]. A recent Surveillance, Epidemiology, and End Results Program analysis noted increased rates of 5-year survival for patients with HNSCC, from 55% to 66%, when comparing outcomes in 1992–1996 vs. 2002–2006. This improvement was mainly attributed to better treatment options and increased survival in patients with HPV-positive HNSCC who tend to be younger and more fit [28].

Significantly higher incidence of HPV-negative HNSCC has been reported for men compared to women (incidence rate 0.19 vs. 0.06 per 1000 person-years, respectively). The imbalance favoring men remains 2.9-fold higher even when comparing subsets of never-smokers/drinkers, which suggests an intrinsic risk factor associated with male sex [29]. Similar results have been observed in other databases [30] and in patients with HPV-positive oropharyngeal disease [31]. Although increased exposure to tobacco and alcohol among men vs. women may partially explain the higher incidence of HPV-negative HNSCC in men, recent understanding of mutational patterns has suggested additional mechanisms that underlie a gender bias in men. Recurrent loss-of-function mutations in X-linked tumor suppressors are observed in HNSCC, including *DDX3X*, *KDM5C*, and *KDM6A* [32]. Mutations in these genes appear more common in men than women with HNSCC. Because men have only one copy of X-linked genes, a single mutation is sufficient to inactivate X-linked tumor suppressor function, thus accelerating transformation. Analysis of the ~800 X-linked coding genes identified 90 genes that modify *TP53*-dependent pathways [33]. Because *TP53* is one of the most commonly mutated genes in HNSCC, X-linked tumor suppressors that modulate *TP53*-dependent pathways present a particular vulnerability for HNSCC transformation in men. Since females have two X chromosomes and males have only one, which is maternally derived, Y-linked effects are more associated with a male bias. Mosaic loss of chromosome Y (LOY) in blood cells is found in approximately a third of men with HNSCC, is disproportionally present among tobacco users, and is associated with shorter survival [34]. Therefore, genomic effects involving recurrent mutations in X-linked tumor suppressors likely contribute to a higher incidence of HPV-negative HNSCC in men, and this bias is not likely only due to greater exposure to tobacco.

## 4. Genomic Instability and Increased Neo-Epitope Burden in Head and Neck Squamous Cell Carcinomas

### 4.1. Frequent Mutations

HNSCCs, especially HPV-negative tumors, are associated with aneuploidy, genomic loss of heterozygosity, and high tumor mutational burden (TMB) [4,35,36]. Median mutational prevalence in HNSCC (mutations per megabase) ranks within the top third of tumors and is associated with mutational spectra related to smoking, APOBEC signatures, ultraviolet damage, and age [35], which are consistent with common exposures associated with HNSCC.

Although HNSCCs are associated with aneuploidy, the observed patterns of mutations are non-random, with recurrent alterations observed at oncogene and tumor suppressor gene loci 3q25-29 (*CCNL1*), 7p12 (*EGFR*), 8q24 (*MYC*), 11q13 (*CCND1*), and 17p13.1 (*TP53*) [4]. Recurrent amplification and deletions result in overexpression of oncogenes or loss of heterozygosity (LOH) of tumor suppressors that regulate key mechanisms associated with cell survival [4,37].

Likewise, although HNSCC is associated with a high TMB, within those variants there is also a series of highly recurrent point mutations. The most frequently mutated genes include *TP53*, *CDKN2A*, *FAT1*, *PIK3CA*, *NOTCH1*, *KMT2D*, and *HRAS* [36,38], with mutations in *TP53* or *CDKN2A* observed in 84% and 59% of patients, respectively.

### 4.2. TP53 Mutations

Mutations in *TP53* are associated with the disruption of numerous homeostatic cellular processes including the maintaining of genomic stability, cell cycle progression, regulating DNA repair mechanisms, and controlling apoptosis [39]. Mutations in *TP53* are a negative prognostic factor in HNSCC and are observed more commonly in (i) non–HPV-associated vs. HPV-positive HNSCC, (ii) in men vs. women, and (iii) in tumors associated with repeated exposure to carcinogens such as those arising in the larynx, tongue, and oral cavity [33,40,41,42]. An analysis of 510 patients with HNSCC (http://gdac.broadinstitute.org/, accessed on 20 August 2022) identified an overall *TP53*-mutation rate of 70.4%, with higher frequencies in tumors of the larynx and hypopharynx (83.5%) and the tongue and oral cavity (75.6%), but much lower frequency in tumors of the oropharynx (28.6%), which tend to be associated with HPV [43]. HNSCC arises via a multistep process involving sub-clonal evolution. *TP53* mutations are often early events in this progression and can be detected in premalignant lesions [44,45]. As with other tumors, most *TP53* mutations are missense mutations and most occur within the DNA-binding domain [43].

*TP53* mutations correlate with specific molecular features including acquired uniparental disomy (aUPD) at chromosome 17p13.3-p13.1 and 3p21.31-p21.1 [46] and a depressed immune signature. Relative to *TP53*-wildtype cases, *TP53*-mutant cancers have been associated with reduced tumor-infiltrating lymphocyte (TIL) markers, including CD8 and natural killer (NK) markers [47]. Likewise, *TP53*-mutated cancers have been associated with a decrease in the ratio of pro- vs. anti-inflammatory cytokines and downregulation of numerous immune activating mechanisms such as antigen processing and presentation [47]. Combined with the observation that patients with *TP53*-mutated HNSCC are found in higher proportions among heavy smokers, which has been shown to dampen immune function, these analyses suggest a strong correlation between *TP53* mutations and tumor-suppressed immunity in HNSCC.

Frequently observed in HNSCC, aUPD may have broader molecular and clinical correlates and implications than just *TP53*. Subsets of HNSCC are beginning to be characterized with unique molecular features. For example, the presence of recurrent aUPD on chromosome 9p correlates with the differential expression of multiple genes (C9orf23, SIGMAR1, and HINT2) and with inferior outcomes, and is more frequently observed in oral cavity cancers than other HNSCCs [37]. Additional studies with larger patient numbers will be required to further refine these molecular subsets, their consequences, and clinical correlates.

A correlation between high levels of genomic damage and decreased disease-free and overall survivals is observed in numerous solid tumors, including HNSCC. Disentangling this effect from *TP53* is challenging because not all *TP53* mutations appear equivalent. Because genome integrity is a *TP53* function, *TP53* mutations diminish DNA repair capacity and enable mutation tolerance, resulting in an accumulation of mutations. High TMB has a negative prognostic significance [48]. Analysis of 1669 HPV-negative HNSCCs found that cases with *TP53* mutations are more likely to be associated with *CDKN2A* mutations or high TMB depending on the tumor subsite. Patterns of recurrent mutations on the *TP53* gene differed depending on whether *CDKN2A* mutations are concurrently present. Missense mutations at hotspot areas near codon 277 are more common in the absence of *CDKN2A* co-mutation, and missense mutations near codons 192 and 157 more common in the presence of *CDKN2A* mutations. In our study, co-occurring mutations were biased toward occurrence at *TP53* amino acid residues that directly contact DNA, whereas *TP53* mutations in the absence of *CDKN2A* mutations were more commonly identified at codons in regions that control TP53 protein stability. A higher frequency of *TP53* truncating mutations associated with total loss of *TP53* function was observed without *CDKN2A* mutations. Furthermore, mean TMB was higher in the presence of damaging loss-of-function or dominant negative *TP53* mutations vs. *TP53* gain-of-function mutations. *TP53* mutations also varied by HNSCC tumor subsite, being highest in laryngeal and oral tumors [49].

These findings indicate that deregulation or loss of normal *TP53* function, the most frequent genomic mutation characterizing HNSCC, may be driving increased TMB that ultimately results in deregulated immune function and impairs the ability to respond to treatment; however, the relationship between the rules of cooperation between these mutations, subsite histology, and clinical outcomes remain incompletely characterized [43,49,50].

## 5. Tumor Mutational Burden and Immune Sculpting

High TMB in HNSCC represents a notable tumor vulnerability. The adaptive immune system interrogates intracellular protein fidelity through the HLA class I pathway. Expressed proteins are digested into 8–10-mer peptides that are presented in the cleft of HLA class I cell-surface proteins [51]. Expressed, non-synonymous mutations may result in altered peptides that are presented by HLA molecules and may be recognized by the adaptive immune system. Therefore, tumors with high TMB are likely to have acquired variants resulting in immunologically active neo-epitopes. Diverse mechanisms that circumvent the presentation of these neo-epitopes have been observed in HNSCC, and many of these have been correlated with inferior clinical outcomes [52,53,54].

In HNSCC, mutations are recurrently observed across multiple genes associated with peptide processing and HLA presentation. In isolation, these mutations are relatively uncommon, but collectively, they represent important and recurrent mechanisms of immune escape. In the Cancer Genome Atlas analysis of HNSCC, *HLA-A* was mutated in only 3% of patients [36]. However, collectively, nearly 20% of patients had somatic abnormalities across diverse antigen processing machinery, including HLA-A, HLA-B, B2M, TAP2, and LMP7 [52].

Immune escape may occur through other mechanisms with clinical implications. Reduced expression of HLA class I genes or peptide processing proteins (e.g., LMP2) enables neo-epitope cloaking and is associated with worse outcomes [54]. Alterations in diverse elements of the HLA peptide processing and presentation machinery have been observed and likely provide mechanisms for immune evasion. Likewise, patients with alterations in the antigen presenting machinery or in HLA class I loci have inferior disease-free survival compared to patients who do not [52].

Altered cytokine profiles in the tumor microenvironment are associated with immune signatures that correlate with prognosis in HNSCC. Changes in immune and chronic inflammatory responses play a critical role in characterizing both tumor aggressiveness as well as response therapies. Both the balance and concentrations of cytokines uniquely regulate inflammation and immune escape mechanisms that often depend on regulatory T cells. For example, IL-6 is a multifunctional and pleiotropic cytokine that is a key regulator of immune response, inflammation, and carcinogenesis in a variety of tumor types. Overexpression of IL-6 and elevated serum and/or saliva concentrations in patients with HNSCC is associated with malignant transformation of premalignant oral lesions, and consistently correlated with poor survival. However, thus far IL-6–targeted therapies have shown only limited benefit as monotherapy [19,55]. IL-10 is an example of an immunosuppressive cytokine that promotes the immune escape by neoplastic cells. Its levels are elevated in tongue leukoplakia tissues with high CD163+ M2 macrophage infiltration in the tumor microenvironment that provides an immunosuppressive microenvironment for tumor growth. [55,56] Infiltration of immune cells, specifically CD8+ TILs, has been associated with improved responses [17,57,58,59]. These results highlight the broad importance of immune escape in HNSCC pathogenesis.

LOH at the HLA locus provides a mechanism to prevent presentation of allele-specific peptides. The class I HLAs consist of three highly variable genes (A, B, and C), each with a polymorphic maternal and paternal allele, providing six available alleles for cell-surface presentation of diverse intracellular peptides. LOH involving the HLA class I locus results in loss of half of the expressed alleles, and therefore a reduction in the number of peptides that may be presented for immune interrogation. NK cells recognize cells that lack expression of HLA [60]; however, cells that have undergone HLA LOH retain expression of cell-surface HLA while removing alleles that may present immunologically active neo-epitopes. This provides a narrow opportunity to evade adaptive immunity without alerting innate immune activation.

Deletion events that lead to LOH are common in cancer, may occur across 20–25% of a tumor’s genome [61], and result in irreversible loss of one of the two involved parental alleles (maternal or paternal). Because LOH events are irreversible, they are maintained throughout subsequent clonal evolution of the tumor [62,63], providing a clonal mark that can be diagnostically or therapeutically relevant. Similar to other immune evasion events, HLA LOH has been associated with adverse outcomes and increased risk of relapse [4].

HLA LOH represents a diagnostic opportunity. HLA LOH has been detected through various research-grade assays such as fluorescence in situ hybridization, single-nucleotide polymorphism arrays, and microsatellite analysis [64]. More recently, standard oncology DNA capture and sequencing platforms have been adopted to detect HLA LOH using NGS assays performed as part of routine diagnostic workups [65,66]. This provides a feasible strategy to integrate HLA LOH into a standard HNSCC diagnostic workup.

## 6. Clinical Implications and Future Direction

Immune evasion is a hallmark of many tumor types, including in HNSCC. Immune evasion occurs as a downstream result of accumulating genetic mutations driven by dysfunction of the tumor suppressor activity and amplification of oncogenes. Therefore, therapies that reestablish recognition and elimination of tumor cells may provide patient-specific therapeutic approaches if tumor-specific targets can be identified. Activity of these types of therapies provides further evidence for the pathologic relevance of neo-epitopes, inducing immune surveillance in HNSCC.

### 6.1. Checkpoint Inhibitors

Checkpoint inhibitors have emerged as a therapeutic strategy that enables immune re-activation to overcome tumor immunosuppressive effects. Upregulation of immune checkpoints on activated T-cells enhances the suppressive function of Tregs and is mediated through interaction with its ligands, programmed death ligand 1 (PD-L1) and programmed death ligand 2, which are expressed on antigen-presenting cells, endothelial and epithelial cells, and activated lymphocytes [67]. Relative to normal T-cells, expression of PD-L1 in the microenvironment and on tumor cells increases in 46% to 100% of patients with HNSCC, depending on the study. This variability is attributed to differences in staining technique and methods of sample preservation. Anatomic variance may also contribute. HNSCC-programmed cell death 1 (PD-1) expression is elevated compared to healthy donors on CD8+ T-cells (mean value 9.5 ± 7.8% vs. 4.5 ± 2.6%) and Treg (mean value 14.5 ± 4.4% vs. 11.3 ± 4.2%), both of which reduced significantly with the addition of nivolumab [68]. CHECKMATE-141 provided randomized validation of nivolumab vs. investigator’s choice chemotherapy, with improved overall survival and HR (7.5 months vs. 5.1 months; HR for death, 0.70; 97.73% CI, 0.51 to 0.96; *p* = 0.01) [69]. Similarly, in the KEYNOTE-040 phase III clinical trial, an advantage in overall survival was demonstrated with pembrolizumab, particularly in the subset of patients with a PD-L1 expression of ≥50% on tumor cells (median overall survival 11.6 months vs. 7.9 months; HR 0.54) [70,71]. Therefore, while checkpoint inhibitors have achieved impressive benefits in response rates, they have not necessarily showed long-term disease control in advanced-stage HNSCC and among patients with relapsed or refractory disease. Targeting immune dysfunction by other means, such as through combination therapy, may result in meaningful improvement in response durability.

Immune checkpoints comprise a broad class of inhibitory molecules. Resistance may emerge through the upregulated expression of alternative immune checkpoints. For example, inhibiting the PD/PD-L1 axis through a PD-1 or a PD-L1 inhibitor results in upregulation of the T-cell immunoglobulin mucin-3 (TIM-3) in PD-1 antibody-bound T-cells, and demonstrates a survival advantage with the addition of a TIM-3 blocking antibody [72]. To bolster anti-tumor activity, checkpoint inhibitors are being explored as part of rational combinations with either traditional chemotherapy or radiotherapy, which may facilitate immune recognition through augmented expression of neoantigens [73,74], or with targeted small molecule tyrosine kinase inhibitors to block growth factor signal transduction pathways. Because high-dose chemotherapy and radiation have been shown to be immunosuppressive, significant effort has been exerted in identifying a minimum dose and an optimum schedule and setting to provide the “priming effect” that is purported to assist the activity of checkpoint inhibitors while dampening the immunosuppressive qualities [67,74]. In the phase III KEYNOTE-048 study, pembrolizumab significantly improved overall survival compared to cetuximab plus platinum-containing chemotherapy in patients with a PD-L1 combined positive score (CPS) of ≥20 and a CPS of ≥1 and achieved a noninferior overall survival in the total population of 882 patients. Pembrolizumab plus chemotherapy also improved overall survival vs. the control arm in patients with a PD-L1 CPS of ≥1, ≥20, and all patients [50]. In a post hoc analysis aiming to further characterize the relationship between PD-L1 expression and survival outcomes, for the patient subset with a PD-L1 CPS between 1 and 19, the addition of pembrolizumab to chemotherapy resulted in a median overall survival of 12.7 months (95% CI, 9.4–15.3) vs. 9.9 months (95% CI, 8.6–11.5) with cetuximab plus chemotherapy (HR, 0.71; 95% CI, 0.54–0.94; *p* = 0.00726). However, for patients with a PD-L1 CPS of <1, survival outcomes with cetuximab plus chemotherapy vs. pembrolizumab plus chemotherapy showed no advantage [75]. In contrast, whether and how to incorporate checkpoint inhibition in the curative intent setting is unclear. Indeed, in the JAVELIN Head and Neck 100 study, the addition of the PD-L1 inhibitor avelumab to definitive chemoradiotherapy followed by adjuvant avelumab failed to meet its primary objective of prolonging progression-free survival [76]. More recently, in the randomized phase III KEYNOTE-412 study, the addition of pembrolizumab concurrent with and after definitive chemoradiotherapy did not improve event-free survival compared to chemoradiotherapy and placebo [77]. Such mixed results suggest that the schedule or setting for the combination may have a meaningful impact on outcomes. For example, combined observations from several small studies have suggested that giving chemotherapy before immunotherapy may lead to better outcomes. Studies are also ongoing for the use of checkpoint inhibitors in the adjuvant setting, after a significant reduction in tumor burden [74] or as maintenance therapy after completion of curative intent, multimodal therapy (NCT03452137) [78].

### 6.2. Combination Approaches

Combination approaches of PD-1 and cytotoxic T lymphocyte-associated protein-4 inhibitors, which showed synergistic efficacy in melanoma [79], were associated with disappointing results in HNSCC [80]. However, more promising outcomes are achieved by partnering checkpoint inhibitors with the anti-EGFR antibody cetuximab, which results in only modest clinical efficacy as a monotherapy in the recurrent or metastatic setting [81]. Preliminary phase II data in the metastatic setting indicate that the combination of cetuximab and pembrolizumab results in a promising overall response rate of 45% [82]. However, in a phase I/II study of concurrent nivolumab and cetuximab in 45 patients, of whom 69% had prior exposure to either cetuximab or a checkpoint inhibitor, 1-year progression-free survival and overall survival rates were only 19% and 44%, respectively. These results suggest that the combination may not improve durability of response, particularly in patients with a history of either therapy [83].

Profound immune dysfunction characterizing HNSCC and recent improvements in understanding of immune evasion mechanisms point to novel molecular targets and pathways that may be explored to reverse the key drivers of malignant pathogenesis. The numerous involved mechanisms may provide insight into the conundrum that despite an improvement in response rates, most combinations have shown only incremental improvements in survival outcomes compared to checkpoint inhibitors alone, particularly in the relapsed/metastatic setting. Preliminary explorations of rational combinations with targeted agents are either ongoing or have shown a limited benefit, and combinations with chemotherapy have shown improvements in response rates albeit inconsistent benefits in response durability and survival outcomes.

### 6.3. Future Directions

Non-structural mechanisms such as epigenetic silencing of B2M, HLA class I, or other antigen-processing genes are common mechanisms of immune evasion and can be seen in nearly three quarters of patients with cancer, including those with HNSCC [53,84]. Nonspecific epigenetic modulators may, therefore, be able to abrogate some of these effects and de-repress expression. In vitro, hypomethylating agents and other DNA methyltransferase inhibitors have been shown to augment the expression of HLA and antigen processing genes [85,86,87]. Members of this class of compounds are approved by the United States Food and Drug Administration (decitabine and azacitidine) for the treatment of myelodysplastic syndrome. Whether these agents could clinically reactivate the immune surveillance of tumors with epigenetic immune escape is not yet known. Likewise, EZH2 and HDAC inhibitors are available and might be able to counter other epigenetic mechanisms of silencing or dysregulating antigen processing in tumor cells. Early single-arm studies have started to test combinations with immunotherapy [88]. Additional studies will be required to determine if and which non-structural events can be easily mitigated. 

Antigen presentation is strongly regulated by interferons and other inflammatory cytokines. Defects in INF signaling and STAT1 signal transduction have been described in multiple cancers [89,90,91]. Chemokines such as CXCL14 have also been implicated in HLA class I expression, especially in HPV-positive HNSCCs [92]. Concurrent chemotherapy and radiation provide an inflammatory milieu that results in the activation of broad cytokines, potentially leading to augmented or restored antigen presentation. Ongoing studies are exploring concurrent radiation and checkpoint inhibitors to determine whether the combination might overcome different mechanisms of immune escape. The addition of other cytokines or chemokines might augment the presentation of neo-epitopes and immune anti-tumor activity.

Defects in cell migration have been implicated in generating a “cold” tumor microenvironment. Tumors with lower CD8+ infiltrating lymphocytes are associated with inferior outcomes [15]. Prior exposure to immunosuppression is associated with inferior outcomes, although mechanisms remain more speculative [93]. HNSCC is also associated with lower peripheral blood CD4+ and CD8+ circulating cells compared to those of healthy individuals, suggesting that broad immune suppression may be contributing to tumorigenesis and not simply mechanisms that protect the local tumor microenvironment from immune infiltration [94]. Monalizumab is an antibody that targets the inhibitory NK group 2A receptor on NK cells and is being tested in HNSCC as a mechanism for altering the cellular microenvironment of the tumor. Initial studies as monotherapy have yielded encouraging responses, but combination approaches may enable better immune reactivation and activity [95].

HLA LOH is an irreversible genomic event that is propagated through subsequent sub-clonal evolution. HLA LOH may provide a means to distinguish tumor from normal tissue in a definitive manner due to this irreversible, clonal loss within tumor cells. A recent strategy has emerged to target these cell-surface epitopes that are absent on malignant cells through use of a logic-gated chimeric antigen receptor (CAR)-T receptor referred to as Tmod [96]. Tmod incorporates an activator with a blocker. The blocker is generated using LIR1 extracellular and intracellular domains and fuses this to an scFv domain targeting HLA-A*02, although additional haplotypes may also be targeted. This provides an inhibitory signal when the CAR-T cell engages a cell with HLA-A*02 expression (i.e., a non-malignant cell) [96]. Although striking results have emerged using CAR-T strategies in hematopoietic diseases, translating these approaches to HNSCC has been challenging owing to nonspecific tumor targets and expected on-target/off-tumor toxicities. Integrating blocking receptors that recognize loss of HLA expression may enable an improved therapeutic index in patients with tumors that have acquired HLA LOH. This strategy may reduce on-target/off-tumor toxicities and improve the therapeutic index sufficiently to enable effective CAR-T targeting in HNSCC.

## 7. Conclusions

HNSCC represents a diverse set of related cancers that emerge from the mucosal surfaces in the oropharynx, larynx, and paranasal sinuses. Multiple risk factors have emerged such as smoking, alcohol exposure, HPV infections, age, and male sex. The high TMB in HNSCC establishes a background of potential neo-epitopes and expressed mutations presented to the adaptive immune system. HNSCCs engage in divergent means of immune escape involving decreased expression and mutagenesis of key genes involved in peptide processing and HLA-based presentation. Acquisition of immune evasion is associated with adverse prognosis and may therefore be an important prognostic marker. HLA LOH represents an irreversible form of immune evasion, which is tractable for clinical measurement via DNA-based NGS platforms. Methods that can reverse immune evasion (e.g., epigenetic targeted therapies) or exploit irreversible genomic events (e.g., logic-gated CAR-Ts) can potentially transform survival outcomes for these patients and offer a future direction for therapeutic development.

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
