# Peer review of "Genomic Immune Evasion: Diagnostic and Therapeutic Opportunities in Head and Neck Squamous Cell Carcinomas"

_jcm, 2022, doi:10.3390/jcm11247259_

Round 1
Reviewer 1 Report
This review is timely and informative. However, the order of the review would benefit from improved organization by specific topic. A few instances are mentioned in the points below.
Listed below are points that would benefit from attention.
11. Regarding risk. Line 115 states “(LOY) is found in approximately a third of men with HNSCC”… Please revise statement to clarify that mosaic LOY in blood cells was evaluated. LOY has been reported to be associated with higher risk of non-hematological cancer diagnosis (Forsberg Nature Genetics 2014, PMID 24777449), although HNSCC is not specifically mentioned in this study. This information is more aligned with risk than the authors’ statement regarding reduced survival (prognosis). The LOY sentence (lines 115-117) seems misplaced among the X-linked gene alterations. The authors could provide a statement that some chromosome Y genes are homologs to genes on the X chromosome, thus linking the LOY and X-linked gene alterations.
22. Also in the risk section, the authors describe interesting genomic findings associated with X-linked genes. The alterations described are observed in the cancer and may have occurred at any time during the development/progression of the cancer. Therefore, this section describes mechanisms that likely underlie the increased risk to men, possibly in the context of tobacco or tobacco plus alcohol exposure. Rather than “…mutational patterns has suggested alternative explanation”, it seems that sex chromosome mutation patterns suggest specific genetic vulnerabilities for men.
33. Frequent mutations section: the HLA locus is mentioned as recurrently aneuploid. HLA genes are not typical tumor suppressor genes, though classifying HLA genes as tumor suppressors has been proposed. It may be helpful to restrict chromosome regions listed to classic tumor suppressor/oncogene regions and describe HLA aneuploidy in the Immune Sculpting section. Also, it may be helpful to some readers to parse the oncogenes and the tumor suppressor genes.
44. Frequent mutations section: Although HNSCC has relatively high TMB, the majority of mutations are within tumor suppressor genes (TP53, FAT1, CDKN2A, KMT2D) and don’t exhibit recurrent hot spot mutations. Only a few frequently mutated genes exhibit “hotspot” mutations (PIK3CA, HRAS). …”With mutations in TP53 and CDKN2A” should be revised to “in TP53 or CDKN2A”, as the co-occurrence mutation frequency is not presented. This paragraph (lines 135 through 137) should be revised for clarity.
55. TP53 Mutations: TP53 is involved in the proper maintenance of cellular processes including genomic stability etc. Mutations in TP53 are not involved in this… The typo in line 140 should be corrected.
66. Line 149, the base of tongue is in the oropharynx and, therefore redundant in this sentence. If the desire is to describe the region of the oropharynx, please appropriately revise this sentence.
77. Line 153, TP53 mutations are predominantly found within the DNA-binding domain encoded portion of the gene but are found in other regions as well.
88. Line 179 and a few other places uses the term “hotspot” mutations for TP53. TP53 mutations are common in HNSCC. However, even though a few sites are recurrently mutated, TP53 mutations are not generally classified as “hotspots”. It is recommended that language referring to TP53 hotspot mutations should be revised to recurrent mutations. If the authors want to use the term “hotspot” for TP53 mutations, recommend specific citation of ref#47 and state statistical method used to define/identify hotspots.
99. Please indicate reference 47 for language describing co-occurrence of TP53 and CDKN2A mutations, lines 177-189. This reference needs to be provided in the paragraph describing the study results. Frankly, the value of the TP53 mutation-CDKN2A mutation co-occurrence section to the main thrust of the review (anti-tumor immunity) is unclear and could be omitted.
110. The paragraph comprised of lines 221-225 seems misplaced, as it splits the HLA discussion and does not seem to be related to TMB and immune sculpting. This paragraph should be moved and language about the roles of IL-6 and IL-10 should be added or the paragraph should be omitted.
111. Line 271: “and hazard ratio” should be removed or HR provided.
112. The authors should carefully consider the statement in lines 275 – 277, as several studies have shown improved survival of HNSCC advanced disease or recurrent/metastatic HNSCC with immune checkpoint blockade.
113. In conclusions, salivary gland tumors are only very rarely of squamous histology. Recommend removing salivary glands from line 396, as this is the first mention of these cancers in this review.
Author Response
Reviewer 1:
Comments and Suggestions for Authors
This review is timely and informative. However, the order of the review would benefit from improved organization by specific topic. A few instances are mentioned in the points below.
Listed below are points that would benefit from attention.
- Regarding risk. Line 115 states “(LOY) is found in approximately a third of men with HNSCC”… Please revise statement to clarify that mosaic LOY in blood cells was evaluated. LOY has been reported to be associated with higher risk of non-hematological cancer diagnosis (Forsberg Nature Genetics 2014, PMID 24777449), although HNSCC is not specifically mentioned in this study. This information is more aligned with risk than the authors’ statement regarding reduced survival (prognosis). The LOY sentence (lines 115-117) seems misplaced among the X-linked gene alterations. The authors could provide a statement that some chromosome Y genes are homologs to genes on the X chromosome, thus linking the LOY and X-linked gene alterations.
We have edited the paragraph for clarity and addressed both the mosaic nature of blood cells evaluated and discussed the overlap of X and Y-associated genes.
- Also in the risk section, the authors describe interesting genomic findings associated with X-linked genes. The alterations described are observed in the cancer and may have occurred at any time during the development/progression of the cancer. Therefore, this section describes mechanisms that likely underlie the increased risk to men, possibly in the context of tobacco or tobacco plus alcohol exposure. Rather than “…mutational patterns has suggested alternative explanation”, it seems that sex chromosome mutation patterns suggest specific genetic vulnerabilities for men.
We have edited the paragraph to include the possibility that X-linked tumor suppressors in men may have independent effects or may potentiate the effect of other risk factors.
- Frequent mutations section: the HLA locus is mentioned as recurrently aneuploid. HLA genes are not typical tumor suppressor genes, though classifying HLA genes as tumor suppressors has been proposed. It may be helpful to restrict chromosome regions listed to classic tumor suppressor/oncogene regions and describe HLA aneuploidy in the Immune Sculpting section. Also, it may be helpful to some readers to parse the oncogenes and the tumor suppressor genes.
We have edited the paragraph to focus on classical tumor suppressors and have removed discussion here of the HLA locus so as to avoid confusion.
- Frequent mutations section: Although HNSCC has relatively high TMB, the majority of mutations are within tumor suppressor genes (TP53, FAT1, CDKN2A, KMT2D) and don’t exhibit recurrent hot spot mutations. Only a few frequently mutated genes exhibit “hotspot” mutations (PIK3CA, HRAS). …”With mutations in TP53 and CDKN2A” should be revised to “in TP53 or CDKN2A”, as the co-occurrence mutation frequency is not presented. This paragraph (lines 135 through 137) should be revised for clarity.
We have edited this section for clarity and have been more careful with description of “hotspot” variants.
- TP53 Mutations: TP53 is involved in the proper maintenance of cellular processes including genomic stability etc. Mutations in TP53 are not involved in this… The typo in line 140 should be corrected.
We have edited this sentence.
- Line 149, the base of tongue is in the oropharynx and, therefore redundant in this sentence. If the desire is to describe the region of the oropharynx, please appropriately revise this sentence.
We have removed “base of tongue” from the sentence.
- Line 153, TP53 mutations are predominantly found within the DNA-binding domain encoded portion of the gene but are found in other regions as well.
We have edited the sentence to be more clear that “most” mutations in TP53 occur in the DNA-binding domain.
- Line 179 and a few other places uses the term “hotspot” mutations for TP53. TP53 mutations are common in HNSCC. However, even though a few sites are recurrently mutated, TP53 mutations are not generally classified as “hotspots”. It is recommended that language referring to TP53 hotspot mutations should be revised to recurrent mutations. If the authors want to use the term “hotspot” for TP53 mutations, recommend specific citation of ref#47 and state statistical method used to define/identify hotspots.
See above. We have made appropriate edits and included the reference.
- Please indicate reference 47 for language describing co-occurrence of TP53 and CDKN2A mutations, lines 177-189. This reference needs to be provided in the paragraph describing the study results. Frankly, the value of the TP53 mutation-CDKN2A mutation co-occurrence section to the main thrust of the review (anti-tumor immunity) is unclear and could be omitted.
We have made these edits and included the reference.
- The paragraph comprised of lines 221-225 seems misplaced, as it splits the HLA discussion and does not seem to be related to TMB and immune sculpting. This paragraph should be moved and language about the roles of IL-6 and IL-10 should be added or the paragraph should be omitted.
These sentences have been moved and we have included expanded discussion of IL-6 and IL-10.
- Line 271: “and hazard ratio” should be removed or HR provided.
These data have been included.
- The authors should carefully consider the statement in lines 275 – 277, as several studies have shown improved survival of HNSCC advanced disease or recurrent/metastatic HNSCC with immune checkpoint blockade.
This discussion has been altered to better reflect current outcomes with IO therapy.
- In conclusions, salivary gland tumors are only very rarely of squamous histology. Recommend removing salivary glands from line 396, as this is the first mention of these cancers in this review.
We have removed salivary glands from this sentence.
Reviewer 2 Report
This review discussed genomic mechanisms related to immune-mediated pressures and propose the prognostic and therapeutic implications of detectable immune escape mechanisms that drive tumorigenesis and disease progression. The content may partially help to understand the mechanism of immune evasion in HNSCC.
1.The arguments are scattered. For example, part 2&3 seem to be less related to the title.
2.In terms of the title, the content of the article is not comprehensive. Immune evasion is a complicated process involved in multiomics variations and various cells. The author mainly focused on genomic instability and tumor mutation burden. I suggest to narrow down the topic of the title.
3.There are many immune subtypes or immune-related scores which could predict prognosis and therapeutic response (such as PMID:33230435). These should be introduced in introduction. Also, the difference and correlation between immune subtypes or immune scores and gene mutation should be discussed.
4.There are many statistics in the review. I suggest a table to sum up the most important statistics.
5.If possible, add a graph abstract to help the reader to understand the review easily.
Author Response
Reviewer 2:
Comments and Suggestions for Authors
This review discussed genomic mechanisms related to immune-mediated pressures and propose the prognostic and therapeutic implications of detectable immune escape mechanisms that drive tumorigenesis and disease progression. The content may partially help to understand the mechanism of immune evasion in HNSCC.
1.The arguments are scattered. For example, part 2&3 seem to be less related to the title.
As per comment 2, we have adjusted the title to better reflect the discussion.
2.In terms of the title, the content of the article is not comprehensive. Immune evasion is a complicated process involved in multiomics variations and various cells. The author mainly focused on genomic instability and tumor mutation burden. I suggest to narrow down the topic of the title.
We recognize that this article focuses primarily on genomic biomarkers and that other omic platforms have relevance. We have adjusted the title to better reflect the content of the review.
3.There are many immune subtypes or immune-related scores which could predict prognosis and therapeutic response (such as PMID:33230435). These should be introduced in introduction. Also, the difference and correlation between immune subtypes or immune scores and gene mutation should be discussed.
We agree that other immune scores may be relevant. We have included additional discussion of IL6 and IL-10 and have included the suggested reference.
4.There are many statistics in the review. I suggest a table to sum up the most important statistics.
We have struggled to generate a sensible table and have decided to include a graphical abstract instead.
5.If possible, add a graph abstract to help the reader to understand the review easily.
We have included a graphical abstract.
Round 2
Reviewer 1 Report
The authors were responsive to criticism. A few typographical and grammatical errors are present in the revised language.